# Older adults' medical preferences for the end of life: a cross-sectional population-based survey in Switzerland

Sarah Vilpert ,[1] Clément Meier,[1,2] Jeanne Berche,[3] Gian Domenico Borasio,[4] Ralf J. Jox,[4,5] Jürgen Maurer[1]

[1]Faculty of Business and Economics (HEC), University of Lausanne, Lausanne, Switzerland
[2]Faculty of Biology and Medicine, University of Lausanne, Lausanne, Switzerland
[3]Centre for Primary Care and Public Health (Unisanté), University of Lausanne, Lausanne, Switzerland
[4]Palliative and Supportive Care Service, Lausanne University Hospital and University of Lausanne, Lausanne, Switzerland
[5]Institute of Humanities in Medicine, Lausanne University Hospital and University of Lausanne, Lausanne, Switzerland

**Correspondence to**
Dr Sarah Vilpert;
sarah.vilpert@unil.ch

## ABSTRACT

**Objectives** Medical decision-making at the end of life is common and should be as patient-centred as possible. Our study investigates older adults' preferences towards three medical treatments that are frequently included in advance directive forms and their association with social, regional and health characteristics.

**Setting** A cross-sectional study using population-based data of wave 8 (2019/2020) of the Swiss component of the Survey of Health, Ageing and Retirement in Europe.

**Participants** 1430 adults aged 58 years and older living in Switzerland.

**Primary and secondary outcome measures** Three questions on the preferences regarding cardiopulmonary resuscitation (CPR); life-prolonging treatment in case of high risk of permanent mental incapacity; reduced awareness (sedation) to relieve unbearable pain and symptoms. Their associations with individuals' social, regional and health characteristics.

**Results** Most older adults expressed a wish to receive CPR (58.6%) and to forgo life-prolonging treatment in case of permanent mental incapacity (92.2%). Most older adults also indicated that they would accept reduced awareness if necessary to receive effective treatment for pain and distressing symptoms (59.2%). Older adults' treatment preferences for CPR and life-prolonging treatment differed according to sex, age, partnership status, linguistic region and health status, while willingness to accept reduced awareness for effective symptom treatment was more similarly distributed across population groups.

**Conclusions** Simultaneous preferences for CPR and refusal of life-prolonging treatment might appear to be conflicting treatment goals. Considering individuals' values and motivations can help clarify ambivalent treatment decisions. Structured advance care planning processes with trained professionals allows for exploring individuals' motivations and values and helps to identify congruent care and treatment goals.

## INTRODUCTION

Advances in medical technologies and extended life expectancy have changed the context of dying in today's societies[1] and often result in a need to make complex medical decisions at the end of life (EOL).[2 3] EOL decisions may concern pain or symptom management, possibly but not necessarily

### STRENGTHS AND LIMITATIONS OF THIS STUDY

⇒ This study replicates the questions on treatment preferences that are usually part of advance directives.
⇒ This study explores treatment preferences in a national population-based sample of older adults.
⇒ The wording of the questions may not provide clear indications of individuals' actual treatment preferences.

with life-shortening effects, withholding or withdrawing life-supporting treatment (eg, cardiopulmonary resuscitation (CPR), artificial nutrition and hydration, mechanical ventilation, dialysis), and, in countries where it is legal, euthanasia and assisted suicide.[4 5] Following the principle of patient autonomy,[6] EOL decisions are jointly made by the patients and/or their surrogates and clinicians.[7] These decisions ought to be based on considerations of prognosis, benefits and risks of different treatments, and patients' values and care goals among others.[8] Patients commonly opt for or against certain treatments based on their perceived benefits or costs in terms of quality of life and prolongation of life, while physicians typically consider the medical appropriateness of the treatments under consideration.

EOL decisions are most often the result of multiple discussions between medical experts, patients and potentially their families which constitute the basis of a process called advance care planning (ACP), when preparing for a patient health situation that may worsen in the future. The ACP outcomes may be recorded in advance directives (ADs). ACP aims to enable patients to receive medical treatments that are better aligned with their values, care preferences and goals.[9] ACP, thus, contributes to maintaining patients' quality of life throughout their care episodes,[10] and eventually achieve what they may have defined as a 'good death' for

themselves during the ACP process.[11 12] ADs record the medical treatments to which a person consents or refuses. ADs can also be used to designate a healthcare proxy to make medical decisions on behalf of that person in the event of a loss of decision-making capacity.[13] At the EOL, more than half of the patients lose their decision-making capacity.[4] While ACP may not always entirely achieve its primary goals such as a positive impact on patients' quality of life or EOL care consistent with patient preference and goals of care, it often achieves other important secondary objectives such as improving the quality of patient–physician communication and congruence in preferences between patients and their caregivers as well as reducing decisional conflict, increasing preferences for comfort care and encouraging the documentation of ACP and ADs.[14 15] Furthermore, the existence of ADs is associated with more out-of-hospital care, an increased focus on comfort care, and may contribute to reducing overtreatment at the EOL.[16]

In Switzerland, ADs have been legally binding since 2013 and individuals can find numerous AD forms published by medical associations, patient organisations or other private initiatives. Most AD forms include information on medical and health situations in which the wishes expressed in ADs are to be applied (individuals' treatment preferences, organ donation and designated healthcare proxy), while some of them also contain a section about personal values, the meaning of life and quality of life and related fears and expectations.[17] ADs can be completed by individuals alone, although completion with the assistance of trained professionals or healthcare specialists is generally recommended.[18] While many people tend to have the impression that ADs are mainly targeted to older adults and persons with terminal illnesses, AD completion is in fact also recommended for healthy younger adults, due to the broader risks of potential loss of decision-making capacity after, for example, an accident.[19]

Understanding treatment preferences commonly indicated in AD forms may be helpful to assess attitudes towards EOL medical decisions in the general population and give insights into how to best initiate discussion on the topic. Preferences regarding specific life-sustaining treatments,[20–22] general EOL care goals such as priority for length or quality of life,[21 23 24] and palliative sedation therapy[25–29] have been studied in the general population, but mostly in hypothetical contexts of severe physical pain or cognitive decline. To our knowledge, these preferences have not been investigated together in the way in which they might appear in AD forms. We, therefore, consider three specific questions taken from the actual AD form (see online supplemental appendix 1) used until 2022 by the Swiss Medical Association FMH (Foederatio Medicorum Helveticorum), the leading physicians' association in Switzerland, which comprises 42 000 members and over 70 medical organisations in the country. In this study, we examine older adults' preferences for CPR, life-prolonging treatment and reduced awareness (sedation)

as measured by the questions from the aforementioned AD form in a large population survey of adults aged 58 years and older living in Switzerland. We also investigate the association between stated treatment preferences and individuals' social, regional and health characteristics. This information is relevant to understanding how older adults view specific EOL treatments and identifying differences in treatment preferences among the population groups.

## METHODS
We used data from the Survey of Health, Ageing and Retirement in Europe (SHARE), which collects internationally comparable multidisciplinary longitudinal microdata on social, economic and health conditions of European aged 50 and over and their partners.[30] We used a subsample of the Swiss-SHARE Wave 8 (2019/2020) dataset, which included respondents who participated in the internationally harmonised in-person interview and the national paper-and-pencil self-administered EOL questionnaire distributed at the end of the interview in Switzerland.[31 32] The EOL questionnaire was developed by the Swiss SHARE team and palliative care experts of the Lausanne University Hospital and concerns preferences, knowledge, attitudes and behaviours towards EOL care planning. In Wave 8 in Switzerland, 2005 respondents participated in the SHARE main interview and 94.3% of them (1891 individuals) also completed the EOL questionnaire. Since the last sample refreshment in Switzerland was in 2011, only 28 respondents to the Swiss-SHARE wave 8 survey, which had entered the study as partners, were aged 50–57. We, therefore, discard these respondents from our sample and limit our analyses to respondents aged 58 years and older who had no missing values on any of the items used in our analysis. Our selected final analytical sample comprises 1430 respondents (figure 1).

### Measures
#### Outcome variables
We included three questions in our EOL questionnaire, which have been adapted directly from the long version of the AD form that was publicly available for download from the Swiss Medical Association FMH website. The adaptations made for inclusion in our survey mainly consisted of transforming the statements into questions, while keeping the vocabulary and contextual information provided as in the Swiss Medical Association's AD forms.

#### Question 1
Imagine that you experience a cardiac and/or respiratory arrest. In this situation, you wish:
−To be resuscitated (1=accept CPR).
−Not to be resuscitated (0=refuse CPR).

#### Question 2
Imagine that you are incapacitated following an accident, a stroke or a heart attack. After initial emergency

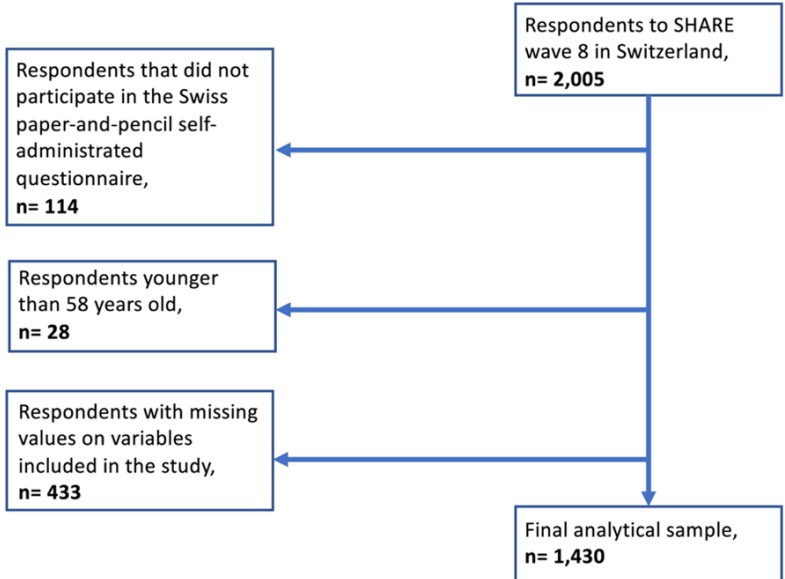

**Figure 1** Flow chart. SHARE, Survey of Health, Ageing and Retirement in Europe.

measures and careful medical assessment, physicians deem it very unlikely that you will regain mental capacity. In this situation, you prefer:

− to forgo all measures which would only serve to prolong your life and suffering (0=refuse life-prolonging treatment)

−that, despite the poor outlook, every medically appropriate measure should be taken (1=accept life-prolonging treatment).

### Question 3

Imagine that you suffer from a disease that causes unbearable pain and symptoms such as fear, restlessness, breathing difficulties and nausea. In this situation:

−you wish to receive optimal treatment of pain and other distressing symptoms, and you are prepared to accept the reduced awareness (sedation) that such treatment may induce (1=accept reduced awareness)

−for you, alertness and the ability to communicate are more important than optimal relief of pain and other symptoms (0=refuse reduced awareness).

### Independent variables

We considered social, regional and health individuals' characteristics as independent variables in our study. Social characteristics included information on sex (0=male, 1=female), age group (58–64 years, 65–74 years, 75+ years), education level based on the International Standard Classification of Education (ISCED) of 2017[33] (basic=ISCED levels 0–2, secondary=ISCED levels 3–4, tertiary=ISCED levels 5–6), living with a partner (0=has a partner, 1=has no partner), parenthood (0=no children, 1=at least one child) and self-perceived financial

difficulties (Is your household able to make ends meet?: 1=easily, 2=fairly easily, 3=with difficulty). Regional characteristics comprised variables on the living environment (0=urban, 1=rural), language regions (German, French or Italian) and religiosity (0=never pray, 1=pray at least sometimes). Health characteristics involved two variables: self-rated health (1=poor/fair health, 2=good health, 3=very good/excellent health) and self-reported limitation in activities of daily living (ADLs) (0=no limitation, 1=at least one limitation).

### Statistical analysis

We performed proportion estimation for all the variables included in the study. We further used probit regressions to assess the partial associations of preferences for medical treatments with respondents' characteristics. Regression estimates are expressed in terms of average partial effects (APEs). APEs report (in percentage points) how the average conditional outcome probability P(Y=1 | X) changes when the given explanatory variable changes from 0 to 1, holding all other independent variables at their observed values. Estimated SEs were clustered at the household level to account for potential unobserved dependencies between partners' responses. All estimations were conducted using STATA/SE V.17.0 software (STATA).

### Patient and public involvement

Patients and the public were not directly involved in the design, conduct, reporting or dissemination of this study

**Table 1** Characteristics of the study sample, adults aged 58+, SHARE Switzerland, 2019/2020, n=1430

| | No | Proportions (%) |
|---|---|---|
| Gender | | |
| Male | 676 | 47.3 |
| Female | 754 | 52.7 |
| Age groups | | |
| 58–64 years | 351 | 24.6 |
| 65–74 years | 603 | 42.2 |
| 75+ years | 476 | 33.3 |
| Education | | |
| Basic | 231 | 16.2 |
| Secondary | 911 | 63.7 |
| Tertiary | 288 | 20.1 |
| Living with a partner | | |
| Yes | 1071 | 74.9 |
| No | 359 | 25.1 |
| Parenthood | | |
| No children | 221 | 15.5 |
| At least on child | 1209 | 84.5 |
| Make ends meet | | |
| Easily | 798 | 55.8 |
| Fairly easily | 457 | 32 |
| With difficulty | 175 | 12.2 |
| Living environment | | |
| Urban | 657 | 45.9 |
| Rural | 773 | 54.1 |
| Switzerland linguistic regions | | |
| German | 1037 | 72.5 |
| French | 345 | 24.1 |
| Italian | 48 | 3.4 |
| Religiosity | | |
| Never pray | 441 | 30.8 |
| Pray at least sometimes | 989 | 69.2 |
| Self-rated health | | |
| Poor/fair health | 250 | 17.5 |
| Good health | 607 | 42.4 |
| Very good/excellent health | 573 | 40.1 |
| Self-reported limitations in activities of daily living | | |
| No limitation | 1343 | 93.9 |
| At least one limitation | 87 | 6.1 |

SHARE, Survey of Health, Ageing and Retirement in Europe.

as it uses SHARE data (secondary data), and direct public contribution was therefore not possible.

## RESULTS

The social, cultural and health characteristics of the study population are presented in table 1. Regarding preferences for medical treatments displayed in figure 2,

58.6% of respondents preferred to receive CPR, 92.2% chose to forgo life-prolonging treatment in case of high risk of permanent mental incapacity and 59.2% accepted reduced awareness for optimum treatment of pain and other distressing symptoms.

Table 2 shows the results from the multivariable probit regressions of preferences for the three different medical treatments on respondents' characteristics. Concerning preferences for CPR, being a woman decreases the probability of wanting to be resuscitated by 11.2 percentage points compared with being a man (APE: −11.2, $p<0.001$). In addition, living without a partner (APE: −8.2; $p<0.05$) and being older than the reference age group of 58–64 (APE 65–74: −10.1, $p<0.01$; APE 75+: −20.4, $p<0.001$) were also each associated with a lower likelihood of wanting to be resuscitated. In contrast, having a tertiary level of education (APE: 10.3, $p<0.05$), being from the Italian-speaking part of Switzerland (APE: 24.8, $p<0.001$), praying at least sometimes (APE: 5.6, $p<0.01$) and self-rating one's health as good (APE: 9.4, $p<0.05$) or very good/excellent (APE: 12.0, $p<0.01$) were positively associated with preferences for CPR.

Regarding general preferences for life-prolonging treatment, being a woman (APE: −4.3, $p<0.01$), being 65–74 years old (APE: −3.8, $p<0.05$) and having at least one limitation in ADL (APE: −4.8, $p<0.05$) were all negatively associated with preference for life-prolonging treatment in case of high risk of permanent mental incapacity. On the other side, living without a partner (APE: 4.3, $p<0.01$) and being from the Italian-speaking part of Switzerland (APE: 17.7, $p<0.01$) were associated with a higher chance of wanting life-prolonging treatment.

Concerning acceptance of reduced awareness, having a secondary level of education (APE: 8.9, $p<0.01$) was associated with a higher likelihood of accepting reduced awareness for optimum pain treatment and other distressing symptoms. At the same time, respondents who said they pray at least sometimes (APE: −6.2, $p<0.01$) were less likely to accept reduced awareness in order to receive optimal treatment of pain and other distressing symptoms.

## DISCUSSION

Our study is the first national population-based study in Switzerland that investigates older adults' preferences towards specific medical treatments that are commonly elicited in official AD forms. Most respondents wished to receive CPR in case of a cardiac/respiratory arrest, to forgo life-prolonging treatment in case of high risk of permanent mental incapacity, and to reduce their awareness for optimal treatment of pain and other distressing symptoms. Additionally, significant social, regional and health-related differences in medical treatment preferences were found for CPR and life-sustaining treatments, while preferences for reduced awareness were more uniformly distributed across different population groups.

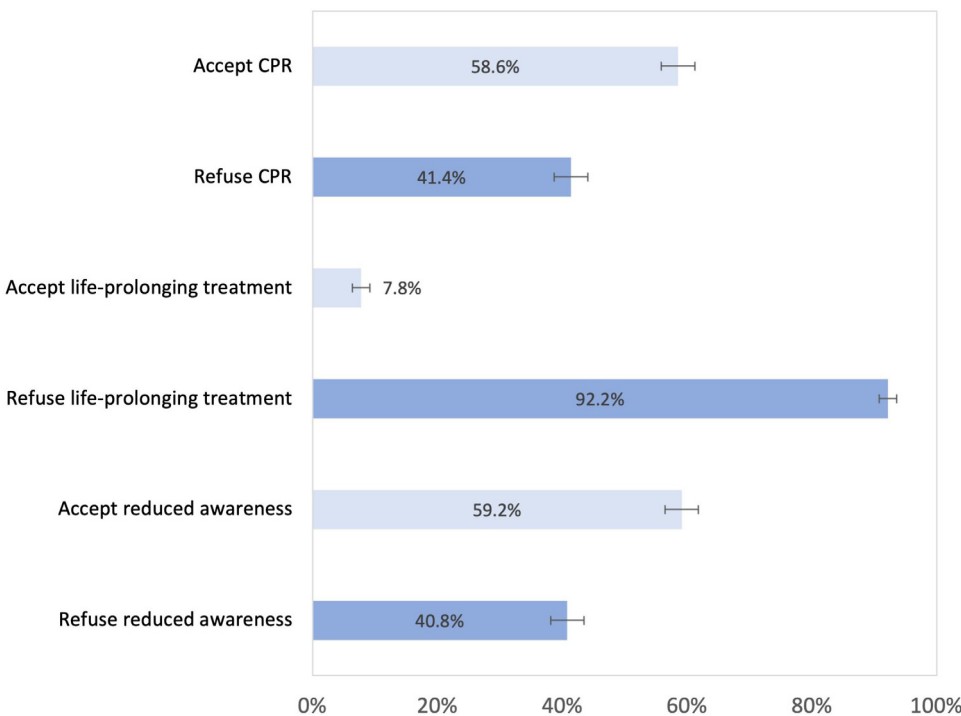

**Figure 2** Preferences for medical treatments, percentages and 95% CIs, adults aged 58+, SHARE Switzerland 2019/2020, n=1430. CPR, cardiopulmonary resuscitation; SHARE, Survey of Health, Ageing and Retirement in Europe.

## Preference for CPR

A majority of individuals reported wanting to receive CPR in the event of cardiac and/or respiratory arrest. In most population-based studies investigating preferences for CPR, the question is asked for a given health condition (eg, terminal cancer, advanced dementia) and individuals largely refuse CPR.[20 21 25] There are several reasons that could explain this difference between our study and previous ones. In our study, the context of the CPR is not specified, and respondents may be referring to their current health status, which is generally rather good (84.8% declared a good, very good or excellent health) with the preference for CPR being potentially influenced by the prospect of significant remaining longevity.[24] Alternatively, this difference could also be explained by an overestimation of CPR survival rate,[34] which may lead individuals to choose CPR.[35] Some individuals may decide against CPR once they have been informed of its procedure, risks and benefits in different contexts.[36]

Regarding social differences in CPR preferences in our study, women reported lower preferences for CPR than men. This finding, which has been shown in other studies,[21 25] could be explained by a higher valuation of quality of life over length of life among women compared with men[37] which may lead them to refrain from treatments.[38] Moreover, women are more likely to have direct experience with caregiving[39] that may make them more aware of the benefits and limitations of certain medical treatments[34] and the potential burden that these treatments may imply for patients and their relatives.[24] Respondents living without a partner were also less likely to opt for CPR than respondents living with a partner.

Since respondents living without a partner also seem to be better informed about the limited success of CPR, as shown in a previous study by the authors based on the same Swiss subsample of SHARE wave 8,[34] they may prefer to refuse CPR.[40]

Conversely, respondents with a tertiary level of education were more likely to report a preference for CPR. Our previous study[34] showed that older adults with higher levels of education were not more knowledgeable about the (relatively low) chances of success of CPR and may, therefore, not be more likely to forgo CPR than persons with less education.[34] A study of hospitalised patients aged 65 and over in New Zealand reported that more educated patients were more likely to prefer CPR in their current state of health.[41] One potential reason could be that individuals with higher education believe to have better health prospects[42] which could induce them to opt for CPR. One study showed that older adults who are optimistic about their own life expectancy are more likely to want to continue all medical treatment in a situation of serious illness even if there are very low chances of survival.[24] Acceptance of CPR was also higher in the Italian linguistic region and among more religious people, who report to pray at least sometimes. These findings may highlight well-documented cultural differences in Switzerland, with respondents from the Italian-speaking part[43 44] and more religious persons being more likely to be influenced by religious considerations and corresponding beliefs in the sanctity of life[38 44 45] that tend to encourage life-prolonging treatments.[11]

Finally, age and self-rated health status, which are two parameters of medical prognoses, were also strongly

**Table 2** Partial associations of preferences for medical treatments with respondents' social, cultural and health characteristics, adults aged 58+, SHARE Switzerland, 2019/2020, n=1430

| | Preference for CPR APE (95% CI) | Preference for life-prolonging treatment APE (95% CI) | Preference for reduced awareness APE (95% CI) |
|---|---|---|---|
| Gender (ref: male) | | | |
| Female | −11.2*** (−16.2, −6.3) | −4.3** (−7.3, −1.4) | 2.3 (−2.9, 7.4) |
| Age groups (ref: 58–64 years) | | | |
| 65–74 years | −10.1** (−16.3, −4.0) | −3.8* (−7.6, −0.1) | 2.2 (−4.2, 8.5) |
| 75+ years | −20.4*** (−27.3, −13.5) | −4.0 (−8.0, 0.0) | −5.1 (−12.4, 2.2) |
| Education (ref: basic) | | | |
| Secondary | 1.2 (−6.0, 8.4) | −4.1 (−8.8, 0.7) | 8.9* (1.4, 16.3) |
| Tertiary | 10.3* (1.5, 19.1) | −4.1 (−9.5, 1.4) | 7.6 (−1.8, 16.9) |
| Living with a partner (ref: yes) | | | |
| No | −8.2* (−14.4, −1.9) | 4.3* (0.5, 8.1) | 2.2 (−4.1, 8.5) |
| Parenthood (ref: no children) | | | |
| At least on child | 0.6 (−6.5, 7.7) | −0.6 (−4.5, 3.2) | 3.3 (−3.9, 10.5) |
| Ease of making ends meet (ref: easy) | | | |
| Fairly easily | 1.6 (−4.2, 7.4) | 1.1 (−2.0, 4.3) | −2.4 (−8.4, 3.7) |
| With difficulty | 2.2 (−6.0, 10.3) | 2.9 (−2.2, 8.1) | −0.0 (−8.7, 8.6) |
| Living environment (ref: urban) | | | |
| Rural | 1.4 (−3.8, 6.6) | 0.3 (−2.5, 3.0) | −0.0 (−5.4, 5.3) |
| Switzerland linguistic regions (ref: German) | | | |
| French | −6.1 (−12.3, 0.0) | 2.3 (−1.1, 5.7) | 2.6 (−3.5, 8.7) |
| Italian | 24.8*** (15.2, 34.3) | 17.7* (4.0, 31.4) | −7.5 (−23.5, 8.5) |
| Religiosity (ref: never pray) | | | |
| Pray at least sometimes | 5.6* (0.1, 11.1) | 0.5 (−2.5, 3.5) | −6.2* (-11.8, −0.5) |
| Self-rated health (ref: poor/fair health) | | | |
| Good health | 9.4* (1.8, 16.9) | −2.4 (−6.6, 1.8) | 2.6 (−4.8, 10.0) |
| Very good/excellent health | 12.0** (4.0, 20.0) | −1.9 (−6.3, 2.5) | −3.3 (−11.4, 4.7) |
| Self-reported limitations in activities of daily living (ref: no limitation) | | | |
| At least one limitation | −1.7 (−12.7, 9.4) | −4.8* (−8.6, −0.9) | 0.9 (−10.4, 12.3) |
| Observations | 1430 | 1430 | 1430 |

APEs based on probit regression models. All probabilities are multiplied by 100 to express APEs in percentage points. Asterisks indicate levels of significance: ***<0.1%, ** 1%, *5%. Interpretation of APEs: Being a woman decreases the probability of wanting to be resuscitated by 11.2 percentage points compared with being a man. For variables with three response categories the interpretation is as follows. Being aged 75 and over reduces the probability of wanting to be resuscitated by 20.4 percentage points in comparison with being aged 58–64 (reference category). APE, average partial effect; CPR, Cardiopulmonary resuscitation; SHARE, Survey of Health, Ageing and Retirement in Europe.

associated with CPR preferences. Compared with adults aged 58–64, older respondents were less likely to state that they would prefer to receive CPR, with this negative association increasing in age. These results are consistent with other studies showing that older people are more likely to refuse CPR.[21 25 46] Older adults might realise that age decreases their chances of surviving a CPR[47] and, therefore, be more likely to reject this treatment. The same reasoning may also apply to people in poorer health.[48]

### Preference for life-prolonging treatment

Over 90% of our respondents indicated a preference to forgo life-prolonging treatment if they were to become incapacitated following an accident, a stroke, or a heart attack and were very unlikely to regain mental capacity.

Similar results were found in studies of preferences to continue or stop treatment in the presence of serious physical or mental illness in the general population.[21 24] Interestingly, the population groups more likely to forgo life-prolonging treatments are comparable to those who refused CPR, that is, women, older persons and those in poorer health (1+ADLs). These findings are in line with a study of the general Dutch population that showed that women, older people and people with a history of serious illness tend to prioritise quality of life over quantity.[38] In contrast, Swiss participants from the Italian-speaking region who were more likely to prefer CPR were also more likely to prefer life-prolonging treatment. A surprising result for which we have no explanation is the

preference for life-sustaining treatments among people living without a partner, especially since many of them simultaneously indicated to refuse CPR.

## Preference for reduced awareness

In our study, 59.2% of our respondents accepted reduced awareness in a situation where they would suffer from a disease that causes unbearable pain and symptoms such as fear, restlessness, breathing difficulties and nausea, while for the remaining respondents (39.2%), awareness and the ability to communicate was more important than optimal relief of pain and other symptoms. Other studies of individuals' preferences for different types of palliative sedation (eg, intermittent, mild, continuous deep sedation), in different situations and for different durations have produced mixed results, which are not directly comparable to ours.[25–29] Interestingly, although adequate pain and symptoms management is one of the most important features of a good death,[28] preservation of consciousness also plays a central role at the EOL for many people.[11 25 27] In a Dutch study, 61% of adults aged 20+ reported being conscious until death as important. Those individuals were less likely to accept terminal sedation (67%) and the use of high dosages of morphine (67%).[28] These findings highlight the clinical importance to clearly discuss the possible interaction and trade-offs between symptom alleviation and decreased alertness with patients.

Only education and religiosity were associated with individuals' preferences for or against reduced awareness as a side effect of optimal symptom treatment. People with secondary education are more likely to accept reduced awareness, which is consistent with a Belgian study that showed that patients with higher levels of education are more likely to emphasise pain and symptom relief at the EOL.[49] Furthermore, more religious respondents were less likely to accept reduced awareness. Another study pointed that the proportion of individuals rejecting gradual and rapid sedation was higher among more religious people.[29] These associations may be linked to the belief that symptom treatment that reduces awareness also shortens life,[50] especially since religiosity is commonly associated with life-prolonging treatment preferences among older adults.[21 24 38]

## Potential discrepancy in treatment preferences

Our results may at first seem contradictory in that most respondents indicated a preference against life-prolonging treatments but at the same time showed a preference for CPR, which has a relatively low chance of survival. This type of ambivalence has already been noted in a study that showed that about one-third of the Dutch adult population could not make a clear choice between quality and quantity of life,[38] which could be compared in our study with the refusal of EOL treatments (quality of life) and the simultaneous preference for resuscitation (quantity of life). Another study also pointed to inconsistencies between Dutch older adults' general EOL goals

and preferences for specific life-sustaining treatments.[20] Their authors explained these results by the lack of knowledge of the risks and side effects of these treatments in the population.

The apparent contradiction between preferences for CPR and life-prolonging treatments in our study may be related to the framing of our questions. Our CPR question provides no information about the context in which the CPR is to be conducted such as whether the CPR would take place inhospital or outhospital, and individual's health status is also not specified in this question. By contrast, the question on life-prolonging treatment refers to a situation of very bad health which may explain why the overwhelming majority of respondents reported a preference to refuse life-prolonging treatment in this context. Both questions were directly drawn from the Swiss Medical Association's AD form, which was in use at the time of our survey but has been revised since. The long version of this form contains supplementary contextual information on individuals' motivations, health history and social situation, which seems helpful in translating patients' values and preferences into medically actionable information. Nevertheless, this supplementary information may not completely prevent potential conflicting information about individuals' treatment preferences in case of patients' loss of decision capacity, especially if this information may stand in contrast to individuals' treatment wishes as indicated by the check box questions. A broader approach to ACP that can be translated into more nuanced ADs is clearly desirable and increasingly implemented in practice.[10]

## Limitations

One limitation of our study is that our question wording may not be ideal for eliciting people's treatment preferences even if the questions were highly relevant to Swiss EOL policy at the time of our survey due to their close correspondence with a widely known AD form in Switzerland. Another limitation is that people in very poor health or living in a nursing home may be under-represented in our study, despite the considerable efforts of SHARE to interview also sicker people and follow respondents into institutions such as nursing homes. On the other hand, the EOL questionnaire's cooperation rate was high (94.3%), and an analysis of the profile of non-respondents based on individuals' characteristics used in our regression models indicated no obvious under-representation of certain population groups.

## Conclusion

Our study investigates preferences for CPR, life-prolonging treatments and reduced awareness in the general population of adults aged 58 years and older living in Switzerland based on three statements included in the FMH Swiss Medical Association's AD form. Most older adults wished to receive CPR in case of a cardiac/respiratory arrest, forgo life-prolonging treatment in case of high risk of permanent mental incapacity, and reduce

their awareness for optimal treatment of pain and other distressing symptoms. Our study also revealed that individuals' preferences appear to be influenced by characteristics that affect medical prognoses, such as advanced age and poor health. Moreover, women were less likely to report a preference for CPR and life-prolonging treatments than men. Furthermore, the apparent contradiction in our findings between individuals' willingness to receive CPR and their wishes to forgo life-prolonging treatment may reveal knowledge gaps regarding the success chances and side effects of certain treatments and also highlights the potential ambivalence between individuals' EOL care goals and the choice they make for specific EOL treatments. This inherent complexity of making advance decisions in the context of ADs could benefit from further contextual information and elicitation of individuals' values through processes of ACP, which facilitate a clearer understanding of individuals' values, motivations and fears concerning EOL care that can help to better translate ADs and related information into medical decisions at the EOL.

**Contributors** SV designed the study. CM and JB conducted the statistical analysis with the support of SV. SV, CM and JM drafted the manuscript. All authors discussed the interpretation of findings and provided critical revision of the manuscript for important intellectual content. SV is the guarantor of the study.

**Funding** This work was supported by the Swiss National Science Foundation (SNSF), grant number 10001C_188836. This paper uses data from Börsch-Supan, A. (2021). Survey of Health, Ageing and Retirement in Europe (SHARE) Wave 8. Release version: 1.0.0. SHARE-ERIC. Data set. DOI: 10.6103/SHARE.w8.100. The SHARE data collection has been funded by the European Commission, DG RTD through FP5 (QLK6-CT-2001-00360), FP6 (SHARE-I3: RII-CT-2006-062193, COMPARE: CIT5-CT-2005-028857, SHARELIFE: CIT4-CT-2006-028812), FP7 (SHARE-PREP: GA No 211909, SHARE-LEAP: GA No 227822, SHARE M4: GA No 261982, DASISH: GA No 283646) and Horizon 2020 (SHARE-DEV3: GA No 676536, SHARE-COHESION: GA No 870628, SERISS: GA No 654221, SSHOC: GA No 823782) and by DG Employment, Social Affairs & Inclusion through VS 2015/0195, VS 2016/0135, VS 2018/0285, VS 2019/0332 and VS 2020/0313. Additional funding from the German Ministry of Education and Research, the Max Planck Society for the Advancement of Science, the US National Institute on Aging (U01_AG09740-13S2, P01_AG005842, P01_AG08291, P30_AG12815, R21_AG025169, Y1-AG-4553-01, IAG_BSR06-11, OGHA_04-064, HHSN271201300071C, RAG052527A) and from various national funding sources is gratefully acknowledged (see https://share-eric.eu/).

**Competing interests** None declared.

**Patient and public involvement** Patients and/or the public were not involved in the design, or conduct, or reporting, or dissemination plans of this research.

**Patient consent for publication** Not applicable.

**Ethics approval** The study was approved by the research ethics committee of the canton of Vaud in March 2014 with number 66/14. In addition, during data collection, respondents gave their consent to participate in the survey on two occasions, when they agreed over the phone to schedule an interview and at the beginning of the in-person interview.

**Provenance and peer review** Not commissioned; externally peer reviewed.

**Data availability statement** Data are available in a public, wide access repository. SHARE data are publicly available for scientific purposes only through registration on the data distribution platform.

**ORCID iD**
Sarah Vilpert http://orcid.org/0000-0001-8194-0421

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
