## [Reviewer comments · BMJ Open]

ARTICLE DETAILS

TITLE (PROVISIONAL)	Older adults' medical preferences for the end of life: a cross-sectional population-based survey in Switzerland
AUTHORS	Vilpert, Sarah; Meier, Clément; Berche, Jeanne; Borasio, Gian Domenico; Jox, Ralf; Maurer, Jurgen

VERSION 1 – REVIEW

REVIEWER	Fowler , Craig Massey University
REVIEW RETURNED	09-Feb-2023

GENERAL COMMENTS	Thank you for the opportunity to review this study. I found it interesting and useful, but would like to note a few areas for possible clarification. When I refer to page numbers in this review, I am using the 'page x of 19' system (as displayed in the header of the proof). Methods: p.4 (lines 40-45): I don't quite follow this rationale for why only people aged 58+ were included? Why, for instance, are they not representative of the population? What does it mean to enter SHARE as a 'partner'? Relatedly, on p. 5, you explain the approach you took to "account for potential unobserved dependencies between partners' responses." The recruitment section doesn't actually spell out whether people were recruited into the study as couples. If couples were recruited, was this by accident or design? What proportion of your sample were the partners of other people in the study? p. 5 (lines 7-9): You revisit this in the discussion section, but although the second and third vignettes provide necessary context (e.g., in V2 you are "incapacitated" and unlikely to "regain mental capacity", and in V3, you are in "unbearable pain"), vignette 1 seems rather threadbare. I would have thought you would want some sort of additional information (e.g., regarding prognosis) for this to be meaningful. p. 5 (Vignette 3): It's quite hard to work out what the choices presented to participants are with respect to this vignette. Should the slash be placed *after* "for you" instead? And could you make the presentation more straightforward by presenting (in each of the three cases) the contrasting choices on separate lines? p. 5 (line 32): Is there a reason to treat age as an ordinal rather than ratio variable? In my home discipline it would be unusual to present IVs in this sort of list format rather than in a more elaborated paragraph (e.g., "We used several demographic
--

characteristics as independent variables, including gender (coded as 0 = male; 1 = female)..." Incidentally, is there a reason no option for non-binary/trans/intersex was provided?

p. 5 (lines 37-40): As written, options 1, 2, and 3 for 'self-perceived financial difficulties' don't make sense. *What* is easy/fairly easy/with difficulty?

Ethical approval

p. 6, lines 18-20: I apologize in advance because this query might be silly or reflect that as a communication scholar, BMJOpen is not a journal I read routinely. Why is it necessary to state that patients and the public weren't directly involved? And why aren't the participants considered members of the public?

Results

p. 8 (first main paragraph): Purely for the sake of readability, I'd suggest presenting the results pertaining to each of the three vignette outcome preferences in separate paragraphs

p. 8 (lines 4-5): Sorry again if this is a discipline specific thing, but I'm used to p values using decimals rather than commas.

p. 8 (line 8/9): Is it possible to add a brief explanation of how APEs should be interpreted? The previous explanation makes it sound like they could be percentages (and this seems to be confirmed by the note in the table on the following page - I would find it helpful to place the note here instead of in the table). Also, are there any rules of thumb readers should keep in mind for small, medium, large effects (and it's okay if the answer is 'no!')?

p. 8 (line 16). I'd edit this to read "being 65-74 years old" instead of "having 65-74 years old).

Discussion

p. 10 (line 27/27): You write "the context of CPR is not specified, and the respondents certainly refer to their current health status, which is rather good..." Did participants, in fact, refer to their 'good' health status explicitly, or are you offering an inference as to why the proportion of people wanting CPR was so high?

p. 11 (lines 53-55): I appreciate the authors' honesty in saying they can't think of an explanation rather than stretching to make one up!

p. 12 (line 26): It may be beyond the scope of this article, but this framing of a 'good death' is interesting, and I wonder if it might help set up the manuscript nicely if it was introduced earlier. As it stands, the manuscript is very descriptive (which is not a bad thing) - but this more theoretical angle could add some richness. This is no deal-breaker for me, though, and may just reflect my own background and how the discipline-specific journals I'm more familiar with approach the publication process.

p. 12 (lines 50/51): You offer some explanation here for what appear to be somewhat contradictory outcomes. I think, though, that the decontextualization of cardiac arrest from illness/outcomes largely accounts for this. If I'm going to be in unbearable pain, sure---knock me out and let me go! But if I'm in my late 50s and otherwise in good health apart from my heart attack, then damnit, I want you to do everything to bring me back! I thought your findings

	(re what was wanted and what wasn't wanted) seemed really paradoxical at the start of the paper. Having read more of the details, however, the findings seem much less paradoxical! p. 13 (lines 16/17): I totally agree with your point about framing. Arguably, this raises an important question of how outcomes and probabilities/contingencies might be presented to patients in the context of ADs (e.g., with respect to gains and losses). This may be a theme to develop. p. 13 (lines 22/25): Re your second point (about it being inappropriate to learn about the importance of remaining aware for patients who have lost decision-making capacity), I'm not convinced this would, in fact, be inappropriate. I can imagine, for example, not being competent to make decisions yet quite capable of appreciating pleasant sensations. I'm reminded of the well-known Jeremy Bentham quote (admittedly regarding animal rights, but it seems somewhat relevant here): "The question is not, Can they reason?, nor Can they talk? but, Can they suffer?" Clarity of writing On p. 2 (line 24): I'm having a hard time parsing the grammar of this sentence. Is the 'reduced awareness' positioned as a potential circumstance under which they would/wouldn't want resuscitation? Or is this saying that people could express the desire for drugs that would *make* them less aware of what was going on? p. 3 (lines 24/25): I suggest rewording "both goals being difficult to achieve simultaneously" to "with it often being difficult to achieve both goals simultaneously." p. 3 (lines 34/40-ish): currently reads "it is possible to complete a written document...which is commonly referred to as advance directives." I'd the sentence after "make medical decisions on behalf of that patient," then have a new sentence "Such a document is commonly referred to as an advance directive (AD)." p. 4 (line 57): "The adaption mainly consisted in" should read "The adaption mainly consisted of"
--	---

REVIEWER	Escher, Monica University Hospitals Geneva, Division of Palliative Medicine
REVIEW RETURNED	14-Feb-2023

GENERAL COMMENTS	This study examines preferences of participants aged 58 and older about common treatment options found in advance directives (AD) forms, i.e. CPR, life-sustaining treatments when chances of recovering mental capacity are low, and acceptance of sedation for refractory symptoms, and socio-demographic factors associated with preferences. Data are drawn from the 2019/2020 Swiss subset of an international longitudinal survey, the Survey of Health, Ageing, and Retirement in Europe (SHARE), which collects public health and socio-economic data. This is a straightforward study including a large sample size. The statements about treatment options found in an existing commonly used AD form in Switzerland were presented to the participants in
--

paper-and-pencil questionnaire. However, as the authors point out, it is not the best way to elicit preferences. It neither reflects the recommended practice nor were the participants presented with the entire AD form, which begins with a section about the individual's attitudes and experiences about illness and end of life, and quality of life. It is a major concern and it limits the relevance and the interpretation of the results, a limitation not mentioned in the paper. Another concern is the sample included in the study. It is not clear whether participants were representative of the general population aged 58-75 and more.

Title and text: the use of "older adults" is surprising and somewhat confusing. More than half the participants were aged 58 to 64 years, an age group not commonly considered "old" or "older". Moreover the study does not stratify the sample according to age and compare older adults to the age group 58-64 years.

Abstract:

- I. 24, "reduced awareness as a consequence of analgesic medication": this doesn't correspond to the AD form statement.

The wording of the statement is awkward and the authors should comment about it in the text, and explain how they interpreted it for the study. Acceptance of side effects of medication is something different from acceptance of purposeful sedation for refractory symptoms.

- Conclusions, first sentence: preferences can appear contradictory without being contradictory. It may be a lack of clear medical guidance on the individual's part, but it is also a consequence of the design of the study, i.e. survey, single statements without any prior discussion and/or exploration of values.

- Conclusions, second sentence: this statement doesn't relate to the objective and the findings of the study. It is a general statement. Moreover, the authors did not present the entire AD form to the participants, who did not have the opportunity to reflect about their personal values.

- Strengths and limitations: multivariable regression models. The authors performed univariate analysis, but no multivariate analysis of the factors. I wouldn't describe it as a strength, unless it was meant as a limitation.

Introduction: the introduction needs careful revising. It should be shortened and more focused on the topic of the study. The achievements of ADs are overstated. The shortcomings and relatively low rate of completion of ADs internationally led to the development of advance care planning and its documentation, and the utility of ADs have been challenged (see for example Auriemma C, et al: doi:10.1001/jamainternmed.2022.1180). The relevance of the present study should be explained within the widest national and international context of advance care planning. Moreover the introduction contains a number of inaccurate or overly simple statements. Here are a few examples.

- P.3, L.13: nowadays antibiotics are usually not considered a life-sustaining treatment on a level with cardiopulmonary resuscitation or invasive ventilation

- P.3, L. 15-16, "Following the principle of patient autonomy, patients are expected to make decisions about their treatment based on the information provided by physicians": patients are considered partners of their care and experts in their values. The preferred model is to engage patients in respectful discussions and exchange of information with physicians and to involve families when needed in order for the patient to reach a meaningful decision.

- P.3, L. 17-18: whether or not attempt CPR is not commonly discussed in primary care consultations, and not necessarily in nursing homes
- P.3, L. 22-24: quality of life is not synonymous with comfort care. To say that both goals (prolonging life and quality of life) are difficult to achieve simultaneously is overstated and do not take into account the early integration of palliative care. The aim of early palliative care is to maximize quality of life while offering the patient potentially life prolonging treatments.
- P.3, L. 40-44, and following: the relative failure of ADs to ensure that patients receive end of life care concordant with their wishes has been shown, as well as the difficulty for families to act as healthcare surrogates and to make patient-centred decisions.
- P.4, l. 9 (decontextualized preferences), l.15 (AD form): it is partly true. The first section of the AD form chosen by the authors is about values and preferences (see my comments above).

P.4, l. 15: explain which organizations developed the form, how widely it is used and by whom. Since individuals can find numerous AD forms, they may not use this one, and a large Swiss organization dedicated to elderly people (Pro Senectute) has its own AD form.

P.4, l. 15: Which version of the AD form did the authors use? The version on the website is new and was not available at the time of the survey. The form used could be added as an annexe.

Methods:

It would be useful to have more information about the survey (SHARE): how participants are selected (inclusion characteristics), how much representative of the general population they are, which data are collected, response rate, etc.

Representativity of the Swiss respondents is something of an issue:

- L. 42-43: better explain why respondents aged 50-57 in 2019/20 were not representative of the general population

- Figure 1: 2'005 individuals participated in the survey in Switzerland. How many individuals were solicited to participate and, hence, what was the response rate?

- How characteristics of the participants (table 1) compare with the characteristics of the same age groups in the general population?

- It is surprising that 15.2% of participants only stated that their health was fair or poor and that 94.7% reported no functional limitation while 22% were 75 and older. In the limitations section, the authors refer to attrition problems and to individuals in poor health or living in nursing homes not participating in the survey. These aspects should be clearly stated in the methods and how they affect the representativity of the survey population and the study participants.

The third age group is defined as 75 and older. It is a very broad category, especially in an ageing population such it is in Switzerland. What was the age distribution within this category?

It seems that the end of life questionnaire was added in Switzerland and is not part of SHARE. The questionnaire seems to be very comprehensive and it would be interesting to detail its content.

It is not clear whether the vignettes were part of the questionnaire or an addition to it.

The questionnaire surveyed participants' preferences, knowledge, attitudes and behaviours toward end of life care planning. It would have been of great interest to determine in the present study how

these factors are associated with answers to the vignettes. Can the authors explain why it was not done?

Considering the large sample of participants, a multivariate analysis of the factors associated with participant's preferences would provide valuable and informative data.

Please give more details about when and how the questionnaire / vignettes were administered: during the in-person interviews? If so, where did the interviews take place? Postal surveys? If so, number of recalls, if any?

Results:

p.6, l 29-30: see my comment in "Abstract" about the wording of the AD statement and the manner it is interpreted and used for the present study.

Table 1:

Education: explain what "low", "secondary", "tertiary" refer to Self-perceived financial difficulties: doesn't fit the the options "easily", "fairly easily", "with difficulty"

Discussion: the discussion is quite long and descriptive. It should provide more insight into the significance of the results, and how they relate to the known literature. The discussion should be more focused and shorter. As it is, many hypotheses are proposed that are not sufficiently based on evidence. Here are some examples.

- Preference for CPR:

L 25-26 "in our study, the context of CPR is not specified": this is partially true. The wording of the statement is "Imagine that you experience a cardiac and/or respiratory arrest"; the participants have no reason to think of another state of health than their current one. Although the general population overestimate the success rate of CPR, there is no a priori reason to think that (self-rated) very healthy individuals would renounce CPR. All the more so than half the participants were aged 58 to 64 years. So the description of how overestimation of success is conveyed is not really to the point.

L. 44-47, "[women's] direct experience with caregiving makes them aware of the benefits and limitations of medical treatments and the potential burden of a patient in relatives": this is part of the explanation given for more female participants renouncing CPR. The reference (Carr D, Moorman SM. 2009) does not report about the burden of being the caregiver of a severely ill patient, but about the experience of respondents of the painful death of a loved one ("During his/her last week of life, how much pain did your spouse/parent have?"). Moreover the sentence suggests three different things that are not substantiated: 1) that women have more direct experience with caring for an ill loved than men 2) that caregiving makes them aware of the limitations of treatments, and 3) that experience of caregiving is associated with experiencing the patient as a burden

L. 50-51: respondents without a partner were better informed than participants with a partner and less likely to choose CPR. The choice is not surprising, but how do the authors explain that participants without a partner are better informed?

Moreover the sentence "as shown in a study based on the same population as ours (ref. 36)" is misleading. If I am not mistaken, both studies use data collected in the same individuals from the same survey (wave 8 of the Swiss subset of SHARE). If so, it must be clearly stated.

The same comment holds for the part about level of education and preferences for CPR (end of p 10 and beginning of p 11)

- Preferences for reduced awareness

1§: this paragraph describes the results of a number of studies without providing any insight in what they are important in regard with the present study. The choice of studies is not obvious; some are rather old, and ref. 8 (Takla, et al. 2021) for example is not used.

2§: Discussion of religious behaviour (praying) and preferences about sedation is not sufficiently evidence-based

- Potential discrepancy

L.49-51: the authors interpret the participants' preferences concerning CPR and life-sustaining treatments as ambivalence. In the present study however it may not be ambivalence. The statement about CPR describes the intervention without mentioning the outcome of CPR. The statement about life-sustaining treatments is more precise: "very unlikely that you will regain mental capacity." Furthermore, the text is not neutral since it further reads: In this situation, you prefer: to forgo all measures which would only serve to prolong your life and suffering ". The authors should discuss their findings in light of the wording of questions.

Limitations:

- the first two limitations mentioned do not concern the study, but are a critical appraisal of the AD form. The reflection about the influence of the wording of the form on the participants' answers should be in the discussion section.

- The third limitation refers to the collection of data. As already said, the authors should provide more information about the SHARE survey and about the characteristics of the present study population compared to the general population. "Analysis of the profile of non-respondents indicates that no population group was underrepresented": this part should be in the results and which characteristics of non respondents are known must be indicated.

- Only 15% of the participants rated their health as poor or fair. It is somewhat surprising considering that 22% were 75 or older, and it suggests a bias. The authors should comment on that and compare these figures with figures found in other similar surveys.

- A limitation of the study pertains to the wording of the AD statements. Since the wording is flawed, we cannot be sure that the participants' answers truly reflect their preferences

- Another similar limitation is the fact that participants answered the vignettes only and did not complete the AD section about personal values and views about quality of life, illness and end of life.

Conclusion: the conclusion should be shorter. It contains general statements not directly stemming from the results of the study

References: I suggest the authors revise the references and provide the most up to date and relevant ones.

Language: minor revision is needed

VERSION 1 – AUTHOR RESPONSE

Reviewers' comments on paper "Older adults' medical preferences for the end of life: a cross-sectional population-based survey in Switzerland"

Thank you very much for your positive assessment of our study and your comments to improve it further, which have clearly made our manuscript better. In the following document, we respond to your comments point by point and refer to the corresponding changes in the manuscript with "track changes".

Reviewer: 1

Dr. Craig Fowler , Massey University

Comments to the Author:

Thank you for the opportunity to review this study. I found it interesting and useful, but would like to note a few areas for possible clarification. When I refer to page numbers in this review, I am using the 'page x of 19' system (as displayed in the header of the proof).

Methods:

p.4 (lines 40-45): I don't quite follow this rationale for why only people aged 58+ were included? Why, for instance, are they not representative of the population? What does it mean to enter SHARE as a 'partner'? Relatedly, on p. 5, you explain the approach you took to "account for potential unobserved dependencies between partners' responses." The recruitment section doesn't actually spell out whether people were recruited into the study as couples. If couples were recruited, was this by accident or design? What proportion of your sample were the partners of other people in the study? The SHARE study is a longitudinal study that targets adults aged 50 and older. The SHARE Swiss sample is nationally representative of non-institutionalized individuals of this target population at the time it is drawn or refreshed. Once these persons have been recruited, they are also followed into institutions etc if needed to keep them in the study. At the beginning of the study and at periodical sample refreshment times, a representative sample is drawn at random from a register of all non-institutionalized persons aged 50 years and over who were residents of Switzerland (main respondent). Partners of target respondents who agreed to participate in SHARE were eligible for a SHARE interview, whatever their age, if they were willing to participate. The Swiss sample was refreshed in 2011 for the last time. In 2019/2020 (SHARE wave 8), eight years after the last sample refreshment, people younger than 58 could only participate in the SHARE survey in Switzerland as partners of the main respondent. Thus, the few people younger than 58+ in our data sole came from interviews of partners of SHARE main respondents and were, therefore, not representative for the corresponding age brackets of the population living in Switzerland. In view of this issue and the small number of such younger respondents, we decided to exclude from the analytical sample. In our analytical sample, 1'118 individuals are target respondents and 312 are partners participating in the SHARE study. Finally, our analyses account for possible unobserved dependencies between partners' responses (i.e., adjusted standard errors), as we assume that respondents from the same household do not give completely independent responses and/or are subject to common unobservable influences captured in the error terms of our statistical models.

p. 5 (lines 7-9): You revisit this in the discussion section, but although the second and third vignettes provide necessary context (e.g., in V2 you are "incapacitated" and unlikely to "regain mental capacity", and in V3, you are in "unbearable pain"), vignette 1 seems rather threadbare. I would have thought you would want some sort of additional information (e.g., regarding prognosis) for this to be meaningful.

We agree that Question 1 provides no contextual situation in which to interpret it, unlike Questions 2 and 3, and that this may seem suboptimal. However, as we wanted to remain as close as possible to

the question of the AD form of the Swiss Medical Association FMH), where a corresponding question was asked without context such as the location of a potential cardiac and/or respiratory arrest (e.g. home, hospital) or context regarding the health status of the person under consideration. As a result, we decided to keep this question general to best reflect the way the question is posed in the AD form used by the Swiss Medical Association (FMH) at the time of our survey. This rationale is now better highlighted in the discussion section (see lines 358-378).

p. 5 (Vignette 3): It's quite hard to work out what the choices presented to participants are with respect to this vignette. Should the slash be placed *after* "for you" instead? And could you make the presentation more straightforward by presenting (in each of the three cases) the contrasting choices on separate lines?

We have changed the presentation of the responses, placing them on different lines as suggested. Regarding the use of "for you", it might sound strange at first, but again corresponds to the formulation of the responses in the Swiss Medical Association (FMH) AD form at the time of the survey (see document in Appendix 1).

p. 5 (line 32): Is there a reason to treat age as an ordinal rather than ratio variable? In my home discipline it would be unusual to present IVs in this sort of list format rather than in a more elaborated paragraph (e.g., "We used several demographic characteristics as independent variables, including gender (coded as 0 = male; 1 = female)...". Incidentally, is there a reason no option for non-binary/trans/intersex was provided?

We chose to treat age as an ordinal variable for conceptual reasons as it better allows us to capture potential non-linearities in age. The first age group correspond to the period before retirement. The second age group correspond to the first period after retirement when people are generally still healthy and active. The third age group correspond to the period of age where people experience health impairments. In Switzerland, this period starts at even older age than 75 given the high life expectancy. Indeed, it rather starts at 80 or even 85 years old. However, our sample does not allow making an age group 80+ as the number of individuals of this age is too small to allow stable statistical estimates.

We have presented the IV variables in a synthetic way because, to our view, they are standard variables that not require extended explanations and this presentation saves space. The absence of non-binary/trans/intersex category is simply because the SHARE study to date only allows for the response categories male or female when eliciting respondents' sex, which still corresponds to the current handling of information on sex in official civil registries in Switzerland as well.

p. 5 (lines 37-40): As written, options 1, 2, and 3 for 'self-perceived financial difficulties' don't make sense. *What* is easy/fairly easy/with difficulty?

We agree that the label we are using (self-perceived financial difficulties) does not correspond to the response categories. The original question was: "Thinking of your household's total monthly income, would you say that your household is able to make ends meet..." We have now provided detailed information about what exactly the categories refer to (see lines 175-176).

Ethical approval

p. 6, lines 18-20: I apologize in advance because this query might be silly or reflect that as a communication scholar, BMJOpen is not a journal I read routinely. Why is it necessary to state that patients and the public weren't directly involved? And why aren't the participants considered members of the public?

BMJOpen requires a statement about the involvement of patients or the general public in the study design. As neither patients nor general public were involved to develop our study design, we have written a corresponding statement.

Results

p. 8 (first main paragraph): Purely for the sake of readability, I'd suggest presenting the results pertaining to each of the three vignette outcome preferences in separate paragraphs. We have now split the main paragraph into three shorter paragraphs: one for each preference measure/vignette.

p. 8 (lines 4-5): Sorry again if this is a discipline specific thing, but I'm used to p values using decimals rather than commas.

We have now standardized the use of decimal point throughout the entire paper.

p. 8 (line 8/9): Is it possible to add a brief explanation of how APEs should be interpreted? The previous explanation makes it sound like they could be percentages (and this seems to be confirmed by the note in the table on the following page - I would find it helpful to place the note here instead of in the table). Also, are there any rules of thumb readers should keep in mind for small, medium, large effects (and it's okay if the answer is 'no!')?

We have now provided the interpretation of APEs in the text (see lines 213-215). APEs are percentage point differences.

p. 8 (line 16). I'd edit this to read "being 65-74 years old" instead of "having 65-74 years old).

We have corrected the sentence according to your suggestion.

Discussion

p. 10 (line 27/27): You write "the context of CPR is not specified, and the respondents certainly refer to their current health status, which is rather good..." Did participants, in fact, refer to their 'good' health status explicitly, or are you offering an inference as to why the proportion of people wanting CPR was so high?

Yes, we were offering an inference and made it more explicit by modifying the sentence as follows: "In our study, the context of CPR is not specified. When answering this question, respondents may, for example, be referring to their own current health status, which is generally rather good (84.8% declared a good, very good, or excellent health)".

p. 11 (lines 53-55): I appreciate the authors' honesty in saying they can't think of an explanation rather than stretching to make one up!

Thank you for your comment.

p. 12 (line 26): It may be beyond the scope of this article, but this framing of a 'good death' is interesting, and I wonder if it might help set up the manuscript nicely if it was introduced earlier. As it stands, the manuscript is very descriptive (which is not a bad thing) - but this more theoretical angle could add some richness. This is no deal-breaker for me, though, and may just reflect my own background and how the discipline-specific journals I'm more familiar with approach the publication process.

The notion of "good death" which is the aim of advance care planning (ACP) and advance directives is now introduced in the introduction section, together with ACP (see lines 75-78).

p. 12 (lines 50/51): You offer some explanation here for what appear to be somewhat contradictory outcomes. I think, though, that the decontextualization of cardiac arrest from illness/outcomes largely accounts for this. If I'm going to be in unbearable pain, sure---knock me out and let me go! But if I'm in my late 50s and otherwise in good health apart from my heart attack, then damnit, I want you to do everything to bring me back! I thought your findings (re what was wanted and what wasn't wanted) seemed really paradoxical at the start of the paper. Having read more of the details, however, the findings seem much less paradoxical!

This apparent contradiction between treatment preferences is now more developed and discussed in more detail in the discussion section (see lines 347-378).

p. 13 (lines 16/17): I totally agree with your point about framing. Arguably, this raises an important question of how outcomes and probabilities/contingencies might be presented to patients in the context of ADs (e.g., with respect to gains and losses). This may be a theme to develop. The importance of how information is framed in ADs and the impact it may have on individuals' decisions is now developed in lines 358-366 and 390-392.

p. 13 (lines 22/25): Re your second point (about it being inappropriate to learn about the importance of remaining aware for patients who have lost decision-making capacity), I'm not convinced this would, in fact, be inappropriate. I can imagine, for example, not being competent to make decisions yet quite capable of appreciating pleasant sensations. I'm reminded of the well-known Jeremy Bentham quote (admittedly regarding animal rights, but it seems somewhat relevant here): "The question is not, Can they reason?, nor Can they talk? but, Can they suffer?" We agree and we have removed this statement from the limitation section.

Clarity of writing

On p. 2 (line 24): I'm having a hard time parsing the grammar of this sentence. Is the 'reduced awareness' positioned as a potential circumstance under which they would/wouldn't want resuscitation? Or is this saying that people could express the desire for drugs that would *make* them less aware of what was going on?

We have changed the punctuation and hope it is clearer now that these are preferences for three different treatment situations.

p. 3 (lines 24/25): I suggest rewording "both goals being difficult to achieve simultaneously" to "with it often being difficult to achieve both goals simultaneously."

We have changed this sentence according to your suggestion.

p. 3 (lines 34/40-ish): currently reads "it is possible to complete a written document...which is commonly referred to as advance directives." I'd the sentence after "make medical decisions on behalf of that patient," then have a new sentence "Such a document is commonly referred to as an advance directive (AD)."

We have changed this sentence according to your suggestion.

p. 4 (line 57): "The adaption mainly consisted in" should read "The adaption mainly consisted of" We have integrated this correction.

Reviewer: 2

Dr. Monica Escher, University Hospitals Geneva Comments to the Author:

This study examines preferences of participants aged 58 and older about common treatment options found in advance directives (AD) forms, i.e. CPR, life-sustaining treatments when chances of recovering mental capacity are low, and acceptance of sedation for refractory symptoms, and socio-demographic factors associated with preferences. Data are drawn from the 2019/2020 Swiss subset of an international longitudinal survey, the Survey of Health, Ageing, and Retirement in Europe (SHARE), which collects public health and socio-economic data.

This is a straightforward study including a large sample size. The statements about treatment options found in an existing commonly used AD form in Switzerland were presented to the participants in paper-and-pencil questionnaire. However, as the authors point out, it is not the best way to elicit preferences. It neither reflects the recommended practice nor were the participants presented with the entire AD form, which begins with a section about the individual's attitudes and experiences about illness and end of life, and quality of life. It is a major concern and it limits the relevance and the interpretation of the results, a limitation not mentioned in the paper. Another concern is the sample

included in the study. It is not clear whether participants were representative of the general population aged 58-75 and more.

We have made deep modifications in our introduction, integrating your general and detailed comments. Your comments helped us to clarify the goals of our study. We think that it is relevant to assess the preferences for medical treatments that appear in the former Swiss Medical Association (FMH) AD form in the general population. We agree that it is not recommended to complete an advance directive without advice or discussion with an appropriate professional. However, the FMH AD form, as well as other forms largely based on it, were and are often still freely available online for completion. This means that some people who complete those forms may do so on their own. In addition, the Swiss Medical Association (FMH) AD short form only included the questions asked to our respondents, albeit in a slightly different manner, without including any information about individuals' values and experience with illness and end of life. Thus, we think that the way the questions were administered to our respondents does not differ completely from what their experience might be if they had completed the Swiss Medical Association (FMH) AD form themselves. Furthermore, in our questionnaire, the three questions about medical treatment preferences come after a set of questions about their end-of-life preferences (values), their personal experience with the end of life, and questions about their competences with end-of-life care decisions. Under these circumstances, questions about treatment preferences were not totally decontextualized. For this reason, we have removed the following sentence from the introduction: "To our knowledge, these preferences have not been investigated together and in a decontextualized manner, as they are presented in AD forms". Furthermore, we have inserted the following sentence in the limitation section that highlights the possibility that our respondents would have selected different responses in the context of an AD form: "One limitation of our study, already mentioned, is the wording of the questions that is not perfect and certainly leads to some bias in the expression of treatment preferences in respondents."

Finally, we provide more information about the population sample in the method section and we answer to each of your important comments below.

Title and text: the use of "older adults" is surprising and somewhat confusing. More than half the participants were aged 58 to 64 years, an age group not commonly considered "old" or "older". Moreover the study does not stratify the sample according to age and compare older adults to the age group 58-64 years.

We use the wording "older adults" by comparison with "younger adults" not included in the SHARE study. This terminology is descriptive and commonly used in the publications using SHARE data and we would, therefore, like to stick to this terminology, not least that a leading alternative "middle-aged and older" often becomes repetitive and unwieldy in the context of a longer paper.

We do not stratify the sample according to age because the differences between age groups are not the focus of this paper. However, we control for and discuss this dimension by introducing age groups into the regression models.

Abstract:

- I. 24, "reduced awareness as a consequence of analgesic medication": this doesn't correspond to the AD form statement. The wording of the statement is awkward and the authors should comment about it in the text, and explain how they interpreted it for the study. Acceptance of side effects of medication is something different from acceptance of purposeful sedation for refractory symptoms. We have changed the sentence to read "reduced awareness (sedation) to relieve unbearable pain and symptoms" (see lines 16-17).

- Conclusions, first sentence: preferences can appear contradictory without being contradictory. It may be a lack of clear medical guidance on the individual's part, but it is also a consequence of the design of the study, i.e. survey, single statements without any prior discussion and/or exploration of values.

We state now that the exploration of patients' values may allow to clarify this apparent contradiction (see lines 25-29).

- Conclusions, second sentence: this statement doesn't relate to the objective and the findings of the study. It is a general statement. Moreover, the authors did not present the entire AD form to the participants, who did not have the opportunity to reflect about their personal values.

The second sentence has now been removed.

- Strengths and limitations: multivariable regression models. The authors performed univariate analysis, but no multivariate analysis of the factors. I wouldn't describe it as a strength, unless it was meant as a limitation.

We replaced it by "Only quantitative data, no text analysis or qualitative study component (mixed methods)".

Introduction: the introduction needs careful revising. It should be shortened and more focused on the topic of the study. The achievements of ADs are overstated. The shortcomings and relatively low rate of completion of ADs internationally led to the development of advance care planning and its documentation, and the utility of ADs have been challenged (see for example Auriemma C, et al: doi:10.1001/jamainternmed.2022.1180). The relevance of the present study should be explained within the widest national and international context of advance care planning. Moreover the introduction contains a number of inaccurate or overly simple statements. Here are a few examples. We have made significant changes to the introduction and incorporated your comments. The limits of ADs and the ACP process are now integrated. In addition, the introduction focuses more on end-of-life decisions and ACP and why it is important to know the preferences for certain treatments in the general population.

- P.3, L.13: nowadays antibiotics are usually not considered a life-sustaining treatment on a level with cardiopulmonary resuscitation or invasive ventilation
We have removed "antibiotics" from the examples.

- P.3, L. 15-16, "Following the principle of patient autonomy, patients are expected to make decisions about their treatment based on the information provided by physicians": patients are considered partners of their care and experts in their values. The preferred model is to engage patients in respectful discussions and exchange of information with physicians and to involve families when needed in order for the patient to reach a meaningful decision.

We have modified the sentence to highlight that patients' medical decision-making is based on a dialogue between patient, relevant family members and physicians (see lines 52-54).

- P.3, L. 17-18: whether or not attempt CPR is not commonly discussed in primary care consultations, and not necessarily in nursing homes
We have removed this sentence.

- P.3, L. 22-24: quality of life is not synonymous with comfort care. To say that both goals (prolonging life and quality of life) are difficult to achieve simultaneously is overstated and do not take into account the early integration of palliative care. The aim of early palliative care is to maximize quality of life while offering the patient potentially life prolonging treatments.

We have completely transformed this paragraph and the decision between quality of life and life prolongation is no longer presented as an exclusive choice.

- P.3, L. 40-44, and following: the relative failure of ADs to ensure that patients receive end of life care concordant with their wishes has been shown, as well as the difficulty for families to act as healthcare surrogates and to make patient-centred decisions.

We have now a paragraph on the goals of ACP, including ADs, and the difficulty of achieving these specific aims. We also mention the positive outcomes of ACP and ADs reported in the literature.

- P.4, l. 9 (decontextualized preferences), l.15 (AD form): it is partly true. The first section of the AD form chosen by the authors is about values and preferences (see my comments above).

P.4, l. 15: explain which organizations developed the form, how widely it is used and by whom. Since individuals can find numerous AD forms, they may not use this one, and a large Swiss organization dedicated to elderly people (Pro Senectute) has its own AD form.

We have now provided more information (where available) on the Swiss Medical Association (FMH) AD form to which our survey questions refer (see lines 111-114).

P.4, l. 15: Which version of the AD form did the authors use? The version on the website is new and was not available at the time of the survey. The form used could be added as an annexe.

We now mention in the paper that our survey questions are based on the old Swiss Medical Association (FMH) AD form (see lines 111-114) and provide it as an appendix.

Methods:

It would be useful to have more information about the survey (SHARE): how participants are selected (inclusion characteristics), how much representative of the general population they are, which data are collected, response rate, etc.

While it is challenging to provide a full summary of a complex study as SHARE in the context of our paper, we aimed to highlight the key aspects needed for our study and provided references to the general study as well (Börsch-Supan et al., 2013).

Representativity of the Swiss respondents is something of an issue:

- L. 42-43: better explain why respondents aged 50-57 in 2019/20 were not representative of the general population

We hope that we have now clarified this point (see lines 131-141).

- Figure 1: 2'005 individuals participated in the survey in Switzerland. How many individuals were solicited to participate and, hence, what was the response rate?

In the SHARE survey, we have different response rate according to the respondents' characteristics. The individual retention rate for the longitudinal sample (respondents who participated in Wave 7 and at least one other previous wave) in wave 8 was 81%. The individual retention rate for the longitudinal sample of panel respondents that did not participate in Wave 7 and any combination of (non-)participation in previous waves, but that were brought back into Wave 8 was 21%. As SHARE is a complex panel study that has different types of respondents it is impossible to simply summarize the response rate. This is the reason why we start Figure 1 with the number of individuals who completed a SHARE interview in wave 8.

- How characteristics of the participants (table 1) compare with the characteristics of the same age groups in the general population?

Information for making this comparison is not available.

- It is surprising that 15.2% of participants only stated that their health was fair or poor and that 94.7% reported no functional limitation while 22% were 75 and older. In the limitations section, the authors refer to attrition problems and to individuals in poor health or living in nursing homes not participating in the survey. These aspects should be clearly stated in the methods and how they affect the representativity of the survey population and the study participants.

The sample issues mentioned in the limitations section are common to population surveys and well described in the literature. Thus, we think that it is sufficient to describe them in the limitation section. However, as there may be further challenges to the full representativeness of our sample, mainly because we have not been able to do a refreshment sample since 2011, we now refer to our sample as a "national population-based sample". This is also the reason why the proportions presented in

Table 1 and in Figure 1 are not weighted anymore: we no longer report to have a representative sample of the population.

The third age group is defined as 75 and older. It is a very broad category, especially in an ageing population such it is in Switzerland. What was the age distribution within this category?

The choice of the age groups is a compromise between theoretical considerations and statistical constraints. The first age group correspond to the period before retirement. The second age group correspond to the first period after retirement when people are generally still healthy and active. The third age group correspond to the period of age where people often experience health impairments. In Switzerland, this period commonly starts at even older age than 75 given the high life expectancy, and may for many people only start at age 80 or even 85 years old. However, generating an age group 80+ would result in a relatively small number of individuals in this age group, making statistical inference considerably more challenging. The distribution in the age category 75+ is the following: Min. 75 years; Max. 99 years; Mean 80.8 years, Median 80 years.

It seems that the end of life questionnaire was added in Switzerland and is not part of SHARE. The questionnaire seems to be very comprehensive and it would be interesting to detail its content. It is not clear whether the vignettes were part of the questionnaire or an addition to it. We specify now that the vignettes were part of the EOL questionnaire (see lines 146-150). We also provide a link where one can find the complete EOL questionnaire (see lines 129-131).

The questionnaire surveyed participants' preferences, knowledge, attitudes and behaviours toward end of life care planning. It would have been of great interest to determine in the present study how these factors are associated with answers to the vignettes. Can the authors explain why it was not done?

This is part of our research agenda for the future. In this first paper on the topic we want to set the landscape on which to build our future research in this domain.

Considering the large sample of participants, a multivariate analysis of the factors associated with participant's preferences would provide valuable and informative data.

As we want to keep the narrative of the paper as straightforward as possible, we believe it is preferable to present only one outcome at a time for ease of interpretation.

Please give more details about when and how the questionnaire / vignettes were administered: during the in-person interviews? If so, where did the interviews take place? Postal surveys? If so, number of recalls, if any?

The EOL questionnaire is a paper-and-pencil self-administrated questionnaire distributed at the end of the in-person interview. Most of the time, the interview takes place at the respondent's home (occasionally it might take place outside the home, but it is not recommended). The interviewers usually wait until the respondents have filled in the self-administrated questionnaire to take it with them. Sometimes respondents may want to complete it on another day. In this case, they have to send them back by mail. We have provided additional information on this in the method section (see lines 126-131).

Results:

p.6, l 29-30: see my comment in "Abstract" about the wording of the AD statement and the manner it is interpreted and used for the present study.

Table 1:

Education: explain what "low", "secondary", "tertiary" refer to Self-perceived financial difficulties: doesn't fit the options "easily", "fairly easily", "with difficulty"

The method section describes how the three education categories are constructed. They are based on the International Standard Classification of Education (ISCED) of 2017 (basic=ISCED levels 0-1-2, secondary=ISCED levels 3-4, tertiary=ISCED levels 5-6).

We have now changed the label of “Self-perceived financial difficulties” for “Make ends meet”. We provide more details in the method section about what belongs to the three education categories (see lines 175-176).

Discussion: the discussion is quite long and descriptive. It should provide more insight into the significance of the results, and how they relate to the known literature. The discussion should be more focused and shorter. As it is, many hypotheses are proposed that are not sufficiently based on evidence. Here are some examples.

- Preference for CPR:

L 25-26 “in our study, the context of CPR is not specified”: this is partially true. The wording of the statement is “Imagine that you experience a cardiac and/or respiratory arrest”; the participants have no reason to think of another state of health than their current one. Although the general population overestimate the success rate of CPR, there is no a priori reason to think that (self-rated) very healthy individuals would renounce CPR. All the more so than half the participants were aged 58 to 64 years. So the description of how overestimation of success is conveyed is not really to the point.

In this paragraph, we try to explain why in our population the proportion of people wanting CPR is higher than in other studies. In our opinion, this proportion may be higher for two reasons: 1) people refer to their current health status (healthy) and consider it worthwhile to request CPR; 2) people are not aware of the CPR procedure and the low chances of survival and consider it worthwhile to request CPR. We believe that it is important to raise these two hypotheses as they are both plausible and we have tried to make the discussion clearer.

L. 44-47, “[women’s] direct experience with caregiving makes them aware of the benefits and limitations of medical treatments and the potential burden of a patient in relatives”: this is part of the explanation given for more female participants renouncing CPR. The reference (Carr D, Moorman SM. 2009) does not report about the burden of being the caregiver of a severely ill patient, but about the experience of respondents of the painful death of a loved one (“During his/her last week of life, how much pain did your spouse/parent have?”).

When quoting (Carr D, Moorman SM. 2009), we were referring to the following paragraph in the discussion section: “Women are more likely than men to reject treatment in the case of cognitive impairment. This finding may reflect the ways that gender shapes family relationships, particularly with respect to caregiving (Bookwala et al., 2001). Women typically have more direct experience with caregiving (Wolff and Kasper, 2006), and may recognize that caring for a person with cognitive impairment is particularly burdensome. Thus, women may prefer to withhold treatment to spare their loved ones from the burden of caring for a cognitively impaired patient”.

Moreover the sentence suggests three different things that are not substantiated: 1) that women have more direct experience with caring for an ill loved than men 2) that caregiving makes them aware of the limitations of treatments, and 3) that experience of caregiving is associated with experiencing the patient as a burden.

You are right the sentence implies the 3 points you mention above (if not in a more nuanced way “potential burden”). It has been reworked to emphasize these three elements (see lines 262-267).

L. 50-51: respondents without a partner were better informed than participants with a partner and less likely to choose CPR. The choice is not surprising, but how do the authors explain that participants without a partner are better informed?

According to the results of Meier et al. (2022), people without a partner seem to be better informed about the chances of CPR success (5 percentage points more likely to give the correct

answer) in comparison to people with a partner. As a result, they may make a more rational decision to avoid CPR because of its rather low survival rate.

Moreover the sentence “as shown in a study based on the same population as ours (ref. 36)” is misleading. If I am not mistaken, both studies use data collected in the same individuals from the same survey (wave 8 of the Swiss subset of SHARE). If so, it must be clearly stated.

Yes, it is correct. We have made this statement more explicit (see lines 270-271).

The same comment holds for the part about level of education and preferences for CPR (end of p 10 and beginning of p 11)

We follow the same reasoning as above: higher awareness of low CPR success rates may conduct to CPR renunciation. So, as we observed (C. Meier et al., 2023) that more educated people are not more aware of low CPR success rates, we do not expect this population to be more likely to not want CPR.

- Preferences for reduced awareness

1§: this paragraph describes the results of a number of studies without providing any insight in what they are important in regard with the present study. The choice of studies is not obvious; some are rather old, and ref. 8 (Takla, et al. 2021) for example is not used.

We have summarised the results of all the studies quoted. Although some of these studies are old, we want to keep them as part of a comprehensive listing of the few existing studies on preferences for palliative sedation in the population. The study of Takla et al. 2021 is now mentioned as well.

2§: Discussion of religious behaviour (praying) and preferences about sedation is not sufficiently evidence-based

We have introduced two references and removed the two sentences about spiritual awareness.

- Potential discrepancy

L.49-51: the authors interpret the participants' preferences concerning CPR and life-sustaining treatments as ambivalence. In the present study however it may not be ambivalence. The statement about CPR describes the intervention without mentioning the outcome of CPR. The statement about life-sustaining treatments is more precise: "very unlikely that you will regain mental capacity."

Furthermore, the text is not neutral since it further reads: In this situation, you prefer: to forgo all measures which would only serve to prolong your life and suffering ". The authors should discuss their findings in light of the wording of questions.

We have now taken out of the limitation section the discussion of the wording of our questions that we are now addressing in this paragraph from the angle that you suggest.

Limitations:

- the first two limitations mentioned do not concern the study, but are a critical appraisal of the AD form. The reflection about the influence of the wording of the form on the participants' answers should be in the discussion section.

Yes, it is in the discussion now.

- The third limitation refers to the collection of data. As already said, the authors should provide more information about the SHARE survey and about the characteristics of the present study population compared to the general population. “Analysis of the profile of non-respondents indicates that no population group was underrepresented”: this part should be in the results and which characteristics of non respondents are known must be indicated.

We have now indicated that non-respondents' characteristics explored are the same as those in the regression models (see lines 396-398).

- Only 15% of the participants rated their health as poor or fair. It is somewhat surprising considering that 22% were 75 or older, and it suggests a bias. The authors should comment on that and compare these figures with figures found in other similar surveys.

If we compare our results with those of the Enquête suisse sur la santé (ESS) 2017 (the most recent one), we observe that the proportion of respondents who declare themselves in good/very good health is not so different from that obtained with SHARE data. In the ESS 2017, the proportion is even higher than in SHARE, this is certainly due to the somewhat different labels of the response categories:

SHARE: Diriez-vous que votre santé est...

1. Excellente 2. Très bonne 3. Bonne 4. Acceptable 5. Médiocre

ESS: Comment est votre état de santé en général ? Est-il...

1 - Très bon 2 – Bon 3 – Moyen 4 – Mauvais 5 - Très mauvais

Santé auto-évaluée et problème de santé de longue durée, en 2017

Population de 15 ans et plus vivant en ménage privé

Source: OFS – Enquête suisse sur la santé (ESS)

© OFS 2018

From: <https://www.bfs.admin.ch/bfs/fr/home/statistiques/sante/etat-sante/general.assetdetail.6466163.html>

- A limitation of the study pertains to the wording of the AD statements. Since the wording is flawed, we cannot be sure that the participants' answers truly reflect their preferences. We have mentioned this limitation (see lines 381-392).

- Another similar limitation is the fact that participants answered the vignettes only and did not complete the AD section about personal values and views about quality of life, illness and end of life. This is now discussed in the "Potential discrepancy" paragraph.

Conclusion: the conclusion should be shorter. It contains general statements not directly stemming from the results of the study

Conclusion has been shortened and is limited to the study results.

References: I suggest the authors revise the references and provide the most up to date and relevant ones.

Language: minor revision is needed

Reviewer: 1

Competing interests of Reviewer: No competing interests

Reviewer: 2

Competing interests of Reviewer: None

VERSION 2 – REVIEW

REVIEWER	Fowler , Craig Massey University
REVIEW RETURNED	23-Jun-2023

GENERAL COMMENTS	I appreciate the authors' attention to the feedback I gave re the original manuscript. They have effectively addressed my concerns (either by adding/editing material, or clarifying ways in which I may have misunderstood something). The few additional comments I have the authors are fairly minor. 1. Is it appropriate to categorize the study as (per the abstract) 'observational'?2. p. 4 line 68: I think this line should be placed in parentheses: 'individuals', treatment preferences, organ donation, and designated healthcare proxy'3. Despite the clarification, I'm still not sure why participants aren't considered to be members of the public. I am, however, quite willing to accept that there are BMJ/disciplinary norms and definitions with which I may be unfamiliar.4. Results are now easier to read and interpret, thank you! The explanation provided of the interpretation of APE values is nice and clear. However, because the explanation emphasize their meaning when the value of the explanatory variable changes from 0 to 1, this leaves some ambiguity (to me at least) as to how they should be interpreted when the variable *isn't* dichotomized. In this regard, there may be occasions (e.g., on p. 9) where it would be helpful to make some of the reference categories more explicit when reporting the APEs. E.g., For a variable that's dichotomized such as gender, this can be inferred easily enough; it's not so instinctive in the case of something like region, however (e.g., is the variation associated with being from the Italian-speaking region a contrast with being from the German-language region? The French-language region? Any region that wasn't Italian-speaking?).
--

REVIEWER	Escher, Monica University Hospitals Geneva, Division of Palliative Medicine
REVIEW RETURNED	01-Jun-2023

GENERAL COMMENTS	The authors considered the comments and revised the manuscript accordingly and/or answered the questions.
---

VERSION 2 – AUTHOR RESPONSE

Reviewer: 2

Dr. Monica Escher, University Hospitals Geneva

Comments to the Author:

The authors considered the comments and revised the manuscript accordingly and/or answered the questions.

Reviewer: 1

Dr. Craig Fowler , Massey University

Comments to the Author:

I appreciate the authors' attention to the feedback I gave re the original manuscript. They have effectively addressed my concerns (either by adding/editing material, or clarifying ways in which I may have misunderstood something).

The few additional comments I have the authors are fairly minor.

1. Is it appropriate to categorize the study as (per the abstract) 'observational'?

We have replaced “observational” with “cross-sectional” to make the description of the study more specific.

2. p. 4 line 68: I think this line should be placed in parentheses: 'individuals', treatment preferences, organ donation, and designated healthcare proxy'

Thank you for your comment, we've made the correction.

3. Despite the clarification, I'm still not sure why participants aren't considered to be members of the public. I am, however, quite willing to accept that there are BMJ/disciplinary norms and definitions with which I may be unfamiliar.

We fully understand your concern. We are providing you with all the information that led to the wording of this sentence. The following box appears on the BMJ Open submission form:

Authors must include a Patient and Public Involvement statement in a subsection within the Methods section of their papers. We define patient and public involvement in research as involvement from patients or members of the public in the design, or conduct, or reporting, or dissemination plans of the research. This is distinct from patients and the public being participants in the research. The statement should make the nature and extent of their involvement in the research clear. Protocol articles should state what plans exist for patient/public involvement. We will still consider your work for publication if patients and the public were not (or will not) be involved but you should state this fact clearly in the Patient and Public Involvement subsection.

So we ticked: It was not appropriate or possible to involve patients or the public in the design, or conduct, or reporting, or dissemination plans of our research

Then we were asked: Please paste the Patient and Public Involvement statement included in your research manuscript here:

This is where the following sentence comes from: “Patients and the public were not directly involved in the design of this study as their contribution would not have been relevant to the research question.”

To make the meaning of the sentence more comprehensible, we modified it as follows: “Patients and the public were not directly involved in the design, conduct, reporting or dissemination of this study, as it uses SHARE data (secondary data) and direct public contribution was therefore not possible.”

4. Results are now easier to read and interpret, thank you! The explanation provided of the interpretation of APE values is nice and clear. However, because the explanation emphasize their meaning when the value of the explanatory variable changes from 0 to 1, this leaves some ambiguity

(to me at least) as to how they should be interpreted when the variable *isn't* dichotomized. In this regard, there may be occasions (e.g., on p. 9) where it would be helpful to make some of the reference categories more explicit when reporting the APEs. E.g., For a variable that's dichotomized such as gender, this can be inferred easily enough; it's not so instinctive in the case of something like region, however (e.g., is the variation associated with being from the Italian-speaking region a contrast with being from the German-language region? The French-language region? Any region that wasn't Italian-speaking?).

We are pleased that the results section is now easier to understand. Results for variables with three response categories are to be interpreted in relation to the reference category (as for two-category variables). To make the interpretation of three-category variables transparent, we have provided two additional examples of interpretation. We did not wish to systematically mention the reference category so as not to make the text too long. The first example can be found in the footnote to table 2, and the second at the beginning of the paragraph describing the results of table 2, in the results section.

Example 1:

“For variables with three response categories the interpretation is as follows. Being aged 75 and over reduces the probability of wanting to be resuscitated by 20.4 percentage points in comparison with being aged 58-64 (reference category).”

Example 2:

“In addition, living without a partner (APE: -8.2; $p < 0.05$) and being older than the reference age group of 58-64 (APE 65-74: -10.1, $p < 0.01$; APE 75+: -20.4, $p < 0.001$) were also each associated with a lower likelihood of wanting to be resuscitated.”